# The Role of Prostaglandin E1 as a Pain Mediator through Facilitation of Hyperpolarization-Activated Cyclic Nucleotide-Gated Channel 2 via the EP2 Receptor in Trigeminal Ganglion Neurons of Mice

**DOI:** 10.3390/ijms222413534

**Published:** 2021-12-16

**Authors:** Jean Kwon, Young In Choi, Hang Joon Jo, Sang Hoon Lee, Han Kyu Lee, Heesoo Kim, Jee Youn Moon, Sung Jun Jung

**Affiliations:** 1Department of Biological Sciences, Columbia University, New York, NY 10027, USA; jk4360@columbia.edu; 2Department of Physiology, College of Medicine, Hanyang University, Seoul 04763, Korea; cyi2012@naver.com (Y.I.C.); hangjoonjo@hanyang.ac.kr (H.J.J.); lee4s2@ucmail.uc.edu (S.H.L.); zaguar@hanyang.ac.kr (H.K.L.); 3Department of Anesthesiology and Pain Medicine, Seoul National University College of Medicine, Seoul 03080, Korea; dami0605@snu.ac.kr

**Keywords:** EP2, HCN2 channel, pain, PGE1, trigeminal ganglion neuron

## Abstract

Cyclooxygenase metabolizes dihomo-γ-linolenic acid and arachidonic acid to form prostaglandin (PG) E, including PGE1 and PGE2, respectively. Although PGE2 is well known to play an important role in the development and maintenance of hyperalgesia and allodynia, the role of PGE1 in pain is unknown. We confirm whether PGE1 induced pain using orofacial pain behavioral test in mice and determine the target molecule of PGE1 in TG neurons with whole-cell patch-clamp and immunohistochemistry. Intradermal injection of PGE1 to the whisker pads of mice induced a reduced threshold, enhancing the excitability of HCN channel-expressing trigeminal ganglion (TG) neurons. The HCN channel-generated inward current (I_h_) was increased by 135.3 ± 4.8% at 100 nM of PGE1 in small- or medium-sized TG, and the action of PGE1 on I_h_ showed a concentration-dependent effect, with a median effective dose (ED_50_) of 29.3 nM. Adenylyl cyclase inhibitor (MDL12330A), 8-bromo-cAMP, and the EP2 receptor antagonist AH6809 inhibited PGE1-induced I_h_. Additionally, PGE1-induced mechanical allodynia was blocked by CsCl and AH6809. PGE1 plays a role in mechanical allodynia through HCN2 channel facilitation via the EP2 receptor in nociceptive neurons, suggesting a potential therapeutic target in that PGE1 could be involved in pain as endogenous substances under inflammatory conditions.

## 1. Introduction

Prostaglandin (PG) E is a family of naturally occurring prostaglandins, which are bioactive lipid autocoids (prostanoids). PGE1 and PGE2, the two types of PGE, are metabolized from two different substrates of the lipid-peroxidizing enzyme cyclooxygenase (COX). PGE1 is synthesized from dihomo-r-linoleic acid (DGLA), while PGE2 is converted from arachidonic acid (AA) desaturated from DGLA [1,2]. The clinical usage of PGE1 focuses on its powerful vasodilatory and anti-inflammatory effects [3,4]. As a vasodilator, a PGE1 analog (alprostadil) is used in maintaining the patent ductus arteriosus in newborns with congenital heart defects and in erectile dysfunction treatments [5]. For its anti-inflammatory effects, PGE1 is used in rat arthritis [6] and lupus models [7]. The related mechanism of action of PGE1 is vasodilation through the accumulation of adenosine 3′,5′-cyclic monophosphate (cAMP) by G protein activation in vascular smooth muscle [3,8] and suppression of tumor necrosis factor-induced inflammation due to inhibition of nuclear factor-κB activation and reactive oxygen species production [9].

PGE2, which controls vascular smooth muscle activity and inflammatory responses [1,10], plays an important role in the development and maintenance of hyperalgesia and allodynia [11,12]. It sensitizes peripheral nociceptive nerves and exaggerates inflammatory and neuropathic pain. PGE2-induced hyperalgesic action is thought to occur through specific G protein-coupled receptors, activation of adenylyl cyclase, and increase of cAMP by stimulating PGE receptors. Recent studies highlight that PGE2 significantly stimulates the production of cAMP via the EP4 receptor, resulting in activation of the hyperpolarization-activated cyclic nucleotide-gated 2 (HCN2) channel in inflammatory pain and that the HCN channel has an important role in mechanical allodynia in neuropathic pain conditions [13,14,15]. These studies revealed that HCN2 channels in dorsal root ganglion (DRG) and trigeminal ganglia (TG) neurons have been shown to be involved in nerve injury pain models and that a specific blocker of the HCN channel, ZD7288, reversed both pain behavior and ectopic discharges in injured nerve fibers.

Although PGE1 and PGE2 both stimulate the cAMP-dependent signaling pathway via the same receptors, EP2 and EP4 [16], it is unclear whether PGE1 induces pain by regulating the excitability of sensory neurons as PGE2 does. One of the side effects of PGE1 used to treat erectile dysfunction is penile pain [17], but it is not known whether PGE1 induces this pain, and if so, what mechanism is underlying. While PGE1 and PGE2 have shown opposite effects on inflammation, there is a possibility that PGE1 directly acts on nociceptive sensory neurons because PGE1 and PGE2 have similar cAMP-dependent signaling. There also is a possibility that the target molecule for PGE1 is the HCN channel, and it is likely to involve cAMP-dependent pathway mediation of the EP2 or EP4 receptor, similar to the mechanisms of PGE2. Here, we investigated whether PGE1 induces pain behavior and how it acts on the HCN channel of TG neurons in mice. This study demonstrates the hyperalgesic mechanism of PGE1, which targets the HCN2 channel and increases the excitability of TG neurons via the EP2 receptor and the cAMP signaling pathway

## 2. Results

### 2.1. PGE1 Induces Mechanical Allodynia and Enhances the Excitability of TG Neurons

We examined whether PGE1 affects orofacial pain behavior in mice. PGE1 was injected subcutaneously into the whisker pad of mice at a dose of 10 ng/10 μL per site, after which swiping behavior was observed for 90 min. Injection of PGE1 to vibrissae produced mechanical allodynia (n = 6) in a dose-dependent manner (10 and 20 ng), and normal behavior was observed after 90 min (Figure 1A). In addition, PGE1 induced a depolarization of about 10 mV (control −54.0 ± 0.9 mV, PGE1 −44.3 ± 1.6 mV, n = 9/11 small or medium-sized TG neurons, Figure 1B) and generated spontaneous robust action potentials (n = 4) in some TG neurons. In the change of excitability by current injection, the firing rates of the action potential were facilitated by PGE1 (control 2.9 ± 0.9 rates, PGE1 13.0 ± 2.5 rates, n = 10, Figure 1C). This result suggests that PGE1 produces pain by increasing the excitability of TG neurons.

### 2.2. PGE1 Increases I_h_ Current in TG Neurons in a Dose-Dependent Manner

The excitability of TG neurons has been shown earlier to be determined by HCN channels, inwardly rectifying K^+^ channels (I_Kir_), and some leaky K^+^ channels, which could contribute to the nociception of the orofacial area. In our previous study, we found that orofacial neuropathic pain could be associated with I_h_ that contributes to ectopic action potential firings in sensory TG neurons [13]. Thus, we investigated whether PGE1 affects the excitability of small- or medium-sized TG neurons by acting on ion channels that affect the resting membrane potential. After eliminating I_Kir_ by 200 µM BaCl_2_, application of 100 nM PGE1 reversibly produced an inward current at a holding potential of −60 mV (54.4 ± 9.7 pA, n = 10/12 neurons, Figure 2A). CsCl (2 mM), a broad HCN channel blocker, reduced the PGE1-induced inward current. Also, PGE1 (100 nM, 3 min) significantly enhanced the amplitude of I_h_ to 135.3 ± 4.8% of control levels (n = 9), which recovered to the control value after washout of PGE1 (Figure 2B). This facilitation of I_h_ by PGE1 was observed repeatedly. Next, we examined whether facilitation of amplitude of I_h_ by PGE1 occurred in a dose-dependent manner (1~1000 nM). The maximal effect of PGE1 was a 145.8 ± 5.3% increase of I_h_ at 1 μM, and the median effective dose (ED_50_) for facilitation of I_h_ by PGE1 was 29.3 nM (Figure 2C). These results suggest that the increased excitability of small- or medium-sized TG neurons in the presence of PGE1 was mediated through activation of the HCN channel.

### 2.3. I-V Relationship of PGE1-Induced I_h_ Facilitation

We next tested whether PGE1 affects the I-V relationship of the HCN channel. A comparison of steady-state I_h_ at the end of hyperpolarizing step pulses (arrowhead) from −50 to −120 mV in increments of 10 mV from a holding potential of −50 mV indicated that PGE1 (100 nM) enhanced the amplitude of I_h_ over the full range of voltages (Figure 3A). The reversal potential was measured by examining the tail currents (arrow) from a holding potential of −120 mV to test potentials (−100 to −50 mV) (Figure 3B). For each test voltage, the average peak tail currents were fitted with a linear regression equation, and the mean reversal potential value (E_rev_) was −41.4 mV, which was similar to the range described for the HCN channel [18].

### 2.4. PGE1-Induced Orofacial Pain Is Associated with the HCN Channel

Because the HCN channel might be involved in PGE1-induced pain, we examined the analgesic effect on PGE1-induced pain after blocking the HCN channel. To investigate the relationship between PGE1-induced pain and the HCN channel, CsCl (1.68 μg/10 μL), an HCN channel blocker, was administered simultaneously with PGE1, and the orofacial pain response of mice was observed over 90 min (n = 6, Figure 4A). The orofacial pain score (30~60 min, Figure 4B) was increased by PGE1 (10 ng, 14.0 ± 0.5%; 20 ng, 16.4 ± 0.9%) compared to that of the control group (10.3 ± 0.2%) and was significantly reduced after treatment of PGE1 with CsCl (PGE1 10 ng, 11.3 ± 0.2%; PGE1 20 ng, 12.2 ± 0.2%). However, CsCl alone had no effect on pain response (*p* = 0.245). This result indicates that PGE1-induced pain is mediated by activation of the HCN channel.

### 2.5. PGE1-Induced I_h_ Facilitation Is Mediated by Adenylyl Cyclase and cAMP via the EP2 Receptor

We next investigated the molecular mechanism for the facilitative action of PGE1 on I_h_. Because PGE1 is mediated by activation of adenylyl cyclase, leading to intracellular accumulation of cAMP [19], enhancement of I_h_ by PGE1 was examined to determine whether the enhancement was related to adenylyl cyclase activation. After pretreatment of the TG neurons for 3 min with a potent nonspecific adenylyl cyclase inhibitor, MDL-12,330 A hydrochloride, the amplitude of I_h_ significantly decreased in comparison to the control (69.7 ± 1.6%, n = 5, Figure 5A). Moreover, I_h_ was not facilitated by PGE1 after application of MDL (63.2 ± 8.9%, n = 5, *p* = 0.586). An increase of cAMP has been known to activate HCN2 and HCN4 channels by directly binding in a reversible way to the intracellular cyclic nucleotide-binding domain [5,18]. To determine whether PGE1-induced I_h_ facilitation was affected by intracellular cAMP, the patch electrodes were filled with an intracellular solution containing the cell-permeable cAMP analog, 8-bromo-cAMP (8-Br-cAMP, 100 µM). In the presence of 8-Br-cAMP, PGE1 did not facilitate I_h_ (94.7 ± 5.6%, n = 5, *p* = 0.297) compared to the control (Figure 5B). These results suggest that the facilitative effect of PGE1 is related to downstream activation of adenylyl cyclase and intracellular accumulation of cAMP, and that the HCN2 channels subtype is the HCN channel most relevant in TG associated with PGE1-induced pain.

### 2.6. PGE1-Induced Facilitation of I_h_ Is Mediated via Activation of the EP2 Receptor, Not the EP4 Receptor

To investigate which EP receptor is responsible for the facilitation of I_h_ by PGE1, we used EP1, EP2, and EP 4 receptor antagonists (SC19220, AH6809, and AH23848, respectively), and the EP3 receptor leading the decreases in cAMP was not tested (Figure 6A). Each specific EP receptor antagonist alone did not have an effect on inward I_h_ currents. Facilitation of I_h_ by PGE1 was not altered by SC19220 or AH6809 (122.1 ± 6.2% and 127.4 ± 6.3%, respectively; n = 5), while AH6809 significantly inhibited the facilitation of I_h_ by PGE1 (108.3 ± 5.2%, n = 5, *p* = 0.25). These results suggest that the facilitation of I_h_ by PGE1 is mediated by adenylyl cyclase and cAMP via the EP2 receptor. Next, we tested whether activation of the EP2 receptor by PGE1 was involved in orofacial pain. AH6809 (10 μg/10 μL per site) was injected simultaneously with PGE1 (20 ng), and the orofacial pain response was observed over 90 min (n = 6, Figure 6B). The increased orofacial pain score by PGE1 (30~60 min) was significantly reduced to 12.2 ± 0.2% from 16.4 ± 0.9% after application of AH6809, which (10.9 ± 0.1%) did not show any difference from the control group (10.3 ± 0.2%). Finally, we investigated the protein expression of the EP2 receptor and the HCN2 channel in the TG using immunohistochemistry (Figure 6C). The EP2 receptor and HCN2 channel were expressed in 38.5% (n = 45) and 68.3% (n = 80) of TG neurons (n = 117), respectively. In 29.1% of TG neurons, co-localization of the EP2 receptor and the HCN2 channel was observed, which accounted for 75.6% of EP2 receptor-positive TG neurons. Taken together, these results suggest that HCN2 channel expression in EP2 receptor-positive TG neurons plays an essential role in PGE1-induced orofacial pain and is mediated by activation of the EP2 receptor.

## 3. Discussion

We demonstrated that PGE1 induced orofacial pain in a dose-dependent manner by facilitation of I_h_ in small- or medium-sized TG neurons. Facilitation of I_h_ by PGE1 was supported by the cAMP-sensitive HCN2 channel, which was mediated by adenylyl cyclase and cAMP via activation of the EP2 receptor.

### 3.1. Hyperalgesic Action of PGE1

PGE1 induces vasodilation by causing accumulation of cAMP through EP2 and EP4 in blood vessels. In addition to its vasotropic effects, we hypothesized that PGE1 has direct effects on nociceptive neurons, similar to PGE2, because they share the same cAMP signaling pathway and receptors. Yanagisawa et al. has demonstrated that PGE1 induces depolarization of the peripheral terminals of nociceptive primary afferent fibers, which is associated with a lowering of the threshold for various noxious stimuli [20]. This finding is consistent with our result showing that injection of PGE1 into the whisker pad induces orofacial pain via depolarization of the resting membrane potential of TG neurons. We also observed the pain behaviors of the hind paw by PGE1 and PGE1-induced depolarization of the DRG neuron in another experiment, which indicates that PGE1 can cause pain not only in the orofacial region but also in other peripheral regions, such as the hind paw. In clinical practice, intravenous or intracavernous injections of PEG1 frequently resulted in transient local pain accompanied by phlebitis and angialgia [20,21]. Together, these results suggest that PGE1 can cause pain by acting directly on the peripheral nervous system. In addition, the vasodilatory effect of PGE1 is known to involve an analgesic action in animal models of diabetic neuropathy and spinal stenosis. Although the mechanism is not clear, continuous epidural injection of lipo-PGE1 (a PGE1 analog) improved pain behavior in a rat spinal stenosis model due to increased blood flow and metabolism in the target lesion [22]. Some reports also suggested that PGE1 improved blood flow dysfunction of nerve roots in other animal studies [23], but further research is needed to determine how PGE1 affects not only the hyperalgesic mechanisms but also the analgesic mechanisms.

### 3.2. The Role of the HCN2 Channel in PGE1-Induced Mechanical Allydonia

How PGE1-EP receptor signaling regulates the excitability of nociceptors in TG neurons and how PGE1 signaling develops nociception and through which targets are questions remaining around the role of PGE1 in nociception. PGE1, which has a cAMP signaling pathway similar to that of PGE2 and stimulates cAMP production more effectively than PGE2 [24,25], has the potential to activate the HCN channel as its target. In sensory neurons, the four subtypes of HCN channels, HCN1–HCN4, can be expressed as different homomers in two main respects [18,19]. First, the activation time constants are in the order of HCN1 < HCN2 < HCN3 < HCN4. Second, the HCN2 and HCN4 channels are modulated by an increase in cAMP, with the midpoint of the voltage-activation curve shifted in the positive direction by 12–20 mV, whereas the HCN1 and HCN3 channels show little sensitivity to cAMP. In TG neurons of all sizes, HCN1, HCN2, and HCN3 isoforms have been reported, but HCN4 was poorly expressed [23,26,27]. Although we did not determine mRNA or protein expression of HCN channel isoforms in this study, analysis of the biophysical properties of I_h_ currents in small- or medium-sized TG neurons support the generation of I_h_ by activation of the HCN2 channel, which is very sensitive to cAMP and has intermediate kinetics. We hypothesize that the HCN2 channel is the specific target molecule of the four isoforms. It has been reported that HCN2 channels in DRG and TG neurons are related closely to neuropathic and inflammatory pain through hyperexcitability of nociceptors [14,26,28]. In their study, Chaplan et al. expressed the importance of HCN2 channels in mechanical pain and spontaneous firings which originated from damaged DRG [29]. We have reported that the HCN2 channel is more important for mechanical allodynia than for thermal hyperalgesia, and it could be targeted by a specific antagonist, ZD7288, for HCN channels [13]. Additionally, previous studies have reported HCN2 channels in IB4-negative, calcitonin gene-related peptides, and substance P containing DRG neurons [30,31] were involved in inflammation-induced TG neuron hyperexcitability [32].

The concentration range of PGE1 that induces pain is an important part of evaluating whether PGE1 acts as a physiologically endogenous pain substance. In this study, the ED_50_ of the I_h_ facilitation effect by PGE1 was ~30 nM, and the concentration mainly used in this experiment was 100 nM. Consistent with previous studies, these concentration ranges are physiologically active in vivo and similar to the concentration ranges of antithrombotic action [33,34]. Moreover, the production of PGE1, which exhibits ~20 times stronger biological properties than PGE2 [4], can be determined according to the COX activity or the DGLA/AA ratio [1,35]. Thus, the effective concentration ranges of PGE1 can induce pain under certain conditions in vivo where COX is activated although it may be affected by washout through the blood flow.

This study did not investigate the possibility that PGE1 could affect ion channels other than the HCN2 channel; however, considering that PGE1-induced pain is blocked by CsCl, it can be estimated that the HCN2 channel plays a major role in PGE1-induced pain. Further studies regarding the effects of PGE1 on other pain-related ion channels and the characteristics and distribution of sensory neurons involved in PGE1-induced pain are needed.

### 3.3. The Identification of E-Type Prostaglandin (EP)-Receptor Involved in PGE1-Induced Pain

Next, we focused on which type of EP receptor of PGE1 was involved and the related molecular mechanism. Molecular cloning studies identified four EP receptor subtypes that mediate the effects of PGE1 and PGE2: EP1, EP2, EP3, and EP4. The EP3 receptor has four isoforms: EP3A, EP3B, EP3C, and EP3D [36,37]. The receptor EP1 signals via the Gq-phospholipase C pathway, whereas the EP2, EP3C, and EP4 subtypes signal predominantly via the Gs-cAMP pathway. It is known that PGE1 induces smooth muscle relaxation by increasing the intracellular concentration of cAMP via EP2 and EP4 receptors [8,32] Also, the cardioprotective effects of PGE1 on an ischemic-reperfused heart resulted from EP3 receptor-mediated inhibition of cardiac L-type Ca^2+^ current via a cAMP-dependent pathway [28]. A recent study reported that the EP4 receptor mediates PGE2-induced excitation via HCN2 channels by activating adenylate cyclase [14]. Other studies have suggested that the EP2 receptor subtype mediated spinal inflammatory hyperalgesia by facilitating HCN2 channels [33,34]. An EP2 blocker, AH6809, inhibited PGE1-induced facilitation of I_h_ and PGE1-induced orofacial pain, but neither an EP1 blocker nor an EP4 blocker had the same effect. In addition, in immunohistochemical analysis, HCN2 channels were observed in 80% of EP2-positive TG cells. These findings indicate that the nociceptive action of PGE1 and PGE1-induced I_h_ facilitation is mediated via the EP2 receptor. Patwardhan and colleagues reported that all EP receptor subtypes (EP1 to EP4) were found in TG neurons [38]. EP1 and EP4 were found in less than 5% of all TG neurons, while EP2 and EP3 were prevalent in small- or medium-sized TG neurons (58 and 53% of total neurons, respectively). These reports support our finding that the EP2 receptor mediates PGE1-induced activation of I_h_ in TG neurons. Hence, as PGE1 and PGE2 have different receptors, EP2 and EP4, respectively, the activity of PGE1 needs to be treated as a different pain mechanism from that of PGE2.

In conclusion, PGE1 plays a role in mechanical allodynia through the facilitation of the HCN2 channel via an EP2 receptor by transmitting a signal to adenylyl cyclase and increasing the intracellular concentration of cAMP in small- or medium-sized TG neurons of mice. This finding suggests a potential therapeutic target in that PGE1, a COX metabolite of DGLA, and PGE2, a COX metabolite of AA, could be involved in pain as endogenous substances under inflammatory conditions.

## 4. Materials and Methods

### 4.1. Animals

All experimental procedures for animals were reviewed and approved by the Institutional Animal Care and Use Committee (IACUC) at the College of Medicine, Hanyang University (IACUC Approval No. HY-IACUC-15-0057). Animal treatments were carried out in strict accordance with the ethical guidelines of the International Association for the Study of Pain for the investigation of experimental pain in conscious animals [39] and the National Institute of Health Guide for the Care and Use of Laboratory Animals. Male C57BL/6 wild-type male mice (OrientBio, Sungnam, Korea) weighing 20–25 g were used in this study. The mice were maintained in a conventional facility with a 12/12-h light/dark cycle and fed food and water ad libitum. The temperature and humidity were 22 ± 1 °C and 60%, respectively. All animals were allowed to habituate to the housing facilities for 1 week before the experiments

### 4.2. Behavioral Study—Orofacial Pain

We used an orofacial pain test in accordance with Krzyzanowska et al. [40]. In this study, we used 0.07 g von Frey hair (North Coast Medical, Inc. Morgan Hill, CA, USA) and all mice were restrained by restrainer (1 × 3.75″) for scoring. All mice had a period of adaptation in a room for 1 h before behavioral experiments. A von Frey hair was applied to the whisker pad and the hair was touched whisker pad at a 90° angle until bent in 12 tries. Each experiment was finished within 10 min. We recorded the score according to described below. Swipe across the face is 1 point. Swipes (3 or more) are 1.5 points. Biting or withdrawal of head by the stimulus is 0.25 points. Then mice were removed from the chambers and given an intradermal injection of various drugs into the whisker pad of mice (10 μL/site). We evaluated the number of pain responses for 90 min at 30 min intervals.

### 4.3. Preparation of Trigeminal Ganglia (TG) Neurons

TG neurons from 6~8 week-old male BL6/C57 mice (20~25 g) were prepared as previously described [13] with a minor modification. Animals were anesthetized with isoflurane, and then decapitated. Briefly, TG neurons prepared in 4 °C Hanks balanced solution (HBSS; Life Technologies, Grand Island, NY, USA) were incubated in 3 mL Dubecco’s modified Eagle’s medium (DMEM; Invitrogen^TM^, Life Technologies, Grand Island, NY, USA) containing collagenase (5 mg/mL; Sigma-Aldrich, St. Louise, MO, USA) and dispase (1 mg/mL; Sigma-Aldrich, St. Louise, MO, USA) at 37 °C for 30 min and subsequently in 2 mL DMEM containing 0.25% trypsin (Sigma-Aldrich, St. Louis, MO, USA) at 37 °C for 8 min. Then, the cells were washed in DMEM with 10% fetal bovine serum, triturated with a flame-polished Pasteur pipette to separate cells and remove processes. Subsequently, the separated cells were centrifuged (1000 RPM, 5 min), resuspended, and placed on poly-D-lysine (Sigma-Aldrich, St. Louise, MO, USA)-coated glass coverslips. The cells were maintained at 37 °C in humidified 5% carbon dioxide-95% air incubator.

### 4.4. Electrophysiological Recordings

Experiments with small or medium-sized TG neurons (<30 μm and 30~40 μm in diameter of somata) were performed using the whole-cell patch-clamp technique one day after incubation. Coverslips with TG neurons were transferred to a 0.3 mL recording chamber and perfused continuously with extracellular solution (140 mM NaCl, 5 mM KCl, 2 mM CaCl_2_, 1 mM MgCl_2_, 10 mM glucose, 10 mM HEPES, pH 7.4) at a rate of 5 mL/min using a gravity-fed perfusion system. The recording was started at least 5 min after obtaining the whole-cell configuration.

Whole-cell patch-clamp recordings from TG neurons were made at room temperature using an EPC 10 USB amplifier and analyzed using Pulse program version 8.67 (both from HEKA Electronik, Lambrecht/Pfalz, Germany). Signals were filtered at 1 kHz and sampled at 4 kHz. The micropipettes were manufactured from borosilicate glass capillaries (Harvard Apparatus Ltd., Kent, UK) using a PC-83 puller (Narishige Co., Tokyo, Japan) and the resistance of the pipettes was 5–6 MΩ when filled with intracellular solution (136 mM K-gluconate, 10 mM NaCl, 1 mM MgCl_2_, 10 mM EGTA, 2 mM Mg-ATP, 0.1 mM Na-GTP, pH 7.4).

To investigate the effect of PGE1 on the resting membrane potential of TG neurons, we recorded the PGE1-induced current in a holding potential of −50 mV without injection of hyperpolarization current. BaCl_2_ (200 μM) was used to eliminate inward rectifier K^+^ currents (I_Kir_). We also used CsCl to find out any Cs-sensitive currents by PGE1. In voltage-clamp experiments, I_h_ was elicited from a holding potential of −50 mV by injection of a 1 s pulse of –110 mV in every 20 s. In current-clamp experiments, action potential firings were generated by a series of depolarizing current pulses (0.5 s in duration) from 100 to 500 pA in 50 pA step increments. Then, we tested whether PGE1 has any effects on the action potential firings in TG neurons.

### 4.5. Immunofluorescent Staining

Immunohistochemistry (IHC) was performed as previously described. Briefly, mice were perfused via the ascending aorta with 1% PBS and 4% paraformaldehyde in 0.1 M phosphate buffer (pH 7.4). Collected TGs were fixed at 4 °C overnight in 4% PFA and then transferred to 30% sucrose in PBS for 48 h. Tissue sections of TG (12 μm thickness) were cut in a cryostat (CM3050s; Leica Microsystems Wetzlar, Germany) and processed for immunofluorescence. The sections were washed 3 times in physiological buffer solution (PBS) and then incubated in a blocking solution containing (5% normal horse serum, 2% bovine serum albumin (BSA), and 0.1% Triton X-100) for 1 h. For double immunofluorescent staining, sections were incubated for washed in PBS and then kept by blocking solution containing 5% normal donkey serum 5% FBS (Invitrogen, Waltham, MA, USA) 5% normal donkey serum (Jackson ImmunoResearch, West Grove, PA, USA), 2% BSA (Sigma-Aldrich, Missouri, MO, USA), 2% fetal bovine serum, and 0.1% Triton X-100 for 1 h at room temperature. The sections were incubated overnight at 4 °C with a mixture of mouse anti-HCN2 antibody (1:200, MA5-27718, Invitrogen, Waltham, MA, USA) and rabbit anti-EP2 (1:200, ab2318, Abcam, Waltham, MA, USA) and were then incubated for 1 h at room temperature with a mixture of FITC-conjugated anti-rabbit IgG antibody and Cy3-conjugated anti-mouse IgG antibody (1:1000; Jackson ImmunoResearch, West Grove, PA, USA). The sections were mounted with VectaShield DAPI (Vector Laboratories, Burlingame, CA, USA). Immunostained slides were examined under a confocal laser scanning (20x, LSM700; Carl Zeiss Microscopy GmbH, Jena, Germany), and images were captured with a high-resolution CCD Spot camera (Diagnostic Instruments Inc., Sterling Heights, MI, USA) and analyzed with ZEN-2013 (Carl Zeiss Microscopy GmbH, Jena, Germany).

### 4.6. Drugs

All chemicals were purchased from Sigma. Prostaglandin E1 (PGE1), cis-N-(2-Phenylcyclopentyl)-azacyclotridec-1-en-2-amine hydrochloride (MDL-12,330A hydrochloride), 2-acetylhydrazide 10(11H)-carboxylic acid (SC19220), 6-isopropoxy-9-oxoxanthene-2-carboxylic acid (AH6809), (4Z)-7-[(rel-1S,2S,5R)-5-((1,1′-Biphenyl-4-yl)methoxy)-2-(4-morpholinyl)-3-oxocyclopentyl]- 4-heptenoic acid hemicalcium salt (AH23848), CsCl, and BaCl_2_ were added to the extracellular solutions or vehicle in in vivo test. 8-bromoadenosine 3′,5′-cyclic monophosphate (sodium salt; 8-Br-cAMP) was infused into the cytosol through the patch pipette.

### 4.7. Statistical Analysis

For the in vitro data sets, unpaired, paired Student’s t-test or Mann-Whitney U test was performed to evaluate the differences using Origin 6.0 (Microcal Software, Northampton, MA, USA) and Excel software (Microsoft Corp., Redmond, WA, USA). Data were are presented as means ± SEM and normalized to the control value. The data obtained from the drug application were conducted with Student’s paired *t*-test or one-way repeated measure ANOVA. A *p* value of <0.05 (one-tailed) was considered statistically significant. The numbers of cells tested are indicated in parentheses where applicable. The data were analyzed and plotted using GraphPad Prism6 (GraphPad Software, Inc, California, CA, USA).

Dose-response analysis was performed with Origin 6.1 software (MicroCal, Northampton, MA, USA). The normalized I_h_ was plotted against the PGE1 concentration and fitted using a standard logistic equation, E = E_max_[C^n^/(C^n^ + ED_50_^n^)], where E is the measured effect of the drug, E_max_ is the maximum effect current, C is the drug concentration, ED_50_ is the PGE1 concentration giving 50% of maximum effect, and n is the Hill coefficient of sigmoidicity.

## Figures and Tables

**Figure 1 ijms-22-13534-f001:**
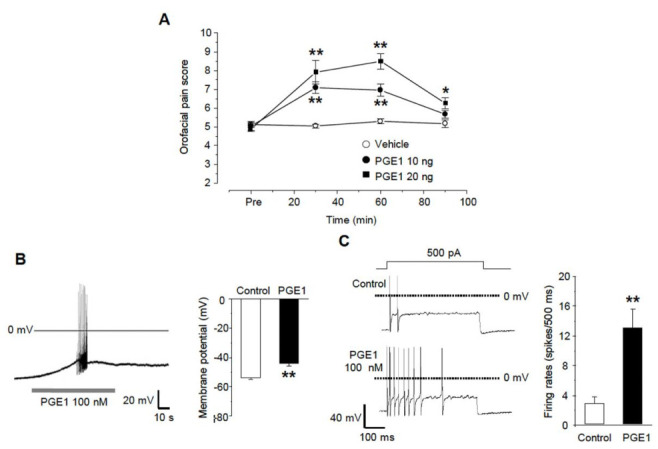
PGE1-induced mechanical allodynia and TG neuronal excitation. (**A**) Orofacial pain score was assessed by pain behavior (swipes and biting or withdrawal of head) during 90 min after the subcutaneous injection of vehicle or PGE1 (10 and 20 ng/10 μL per site) into the whisker pad of mice (score at 30 and 60 min, n = 6, * *p* < 0.05, ** *p* < 0.001 compared to the vehicle group, one-way ANOVA). (**B**) The effects of PGE1 on firing patterns in TG neurons. The application of PGE1 (100 nM) induced the spontaneous bursting action potentials (left) and the depolarization of RMP (right) in current-clamp recording (n = 9, ** *p* < 0.001, paired *t*-test). (**C**) Left: representative traces after a 500 ms current injection (100~500 pA) before and during exposure to PGE1(100 nM) in current-clamp mode. Right: summary of action potential firing rates under each condition (n = 10, ** *p* < 0.001, paired *t*-test).

**Figure 2 ijms-22-13534-f002:**
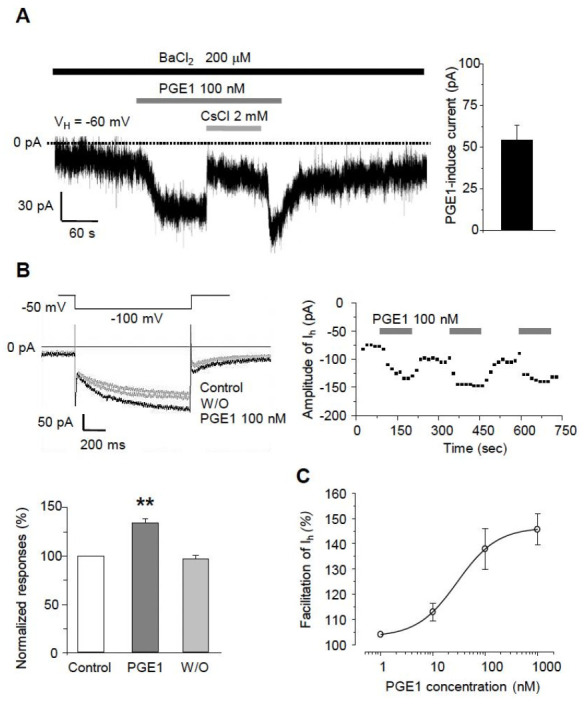
The facilitatory effects of PGE1 on I_h_ in TG neurons. (**A**) Left: Inward currents were activated by PGE1 (100 nM) in the presence of BaCl_2_ at the holding potential of −60 mV in TG neurons. The PGE1-induced inward current was blocked by CsCl (2 mM). Right: Summary of the amplitude of PGE1-induced inward current. (**B**) Recordings of I_h_ were measured during 1 s hyperpolarizing test potential of −110 mV from a holding potential −50 mV with 20 s interval. The traces of I_h_ were superimposed before and after the application of PGE1 (n = 9). The facilitation of I_h_ currents by PGE1 was repetitive. Summary of I_h_ facilitation by PGE1 relative to the control. Results are presented as the mean ± SEM (** *p* < 0.001, paired t-test). (**C**) The concentration-response relationship of facilitation by PGE1. The dose-response curve for PGE1 action on I_h_ in TG neurons. The facilitatory effect of PGE1 (1, 10, 100, and 1000 nM) on I_h_ was in a dose-dependent manner in TG neurons and the median effective dose (ED_50_) of PGE1 was 29.3 nM for I_h_ current. Results are presented as the mean ± SEM.

**Figure 3 ijms-22-13534-f003:**
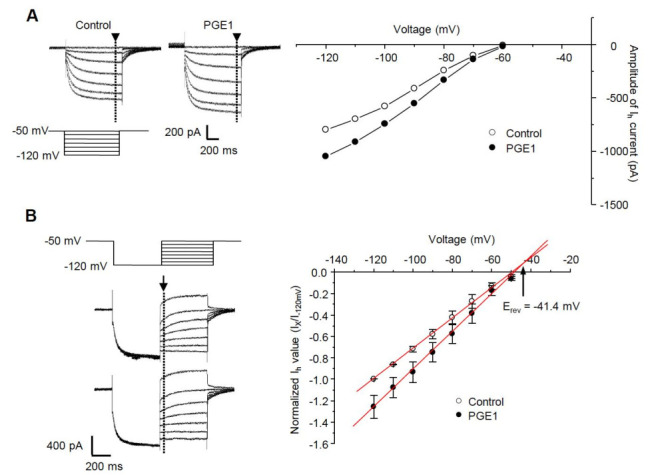
The current-voltage relationship and effects of PGE1 on reversal potential. (**A**) Left: I_h_ was evoked by hyperpolarizing test pulses of −50 mV to −120 mV in 10 mV increments from V_h_ of −50 mV. Representative recordings of I_h_ superimposed before and during exposure to PGE1 (100 nM). Right: Current-voltage relationships were measured at the end of the pulse (arrowhead). (**B**) Left: the reversal potential (E_rev_) was determined by applying a −120 mV hyperpolarizing prepulse for 500 ms then depolarizing in 10 mV increments from −120 mV to −50 mV. Representative recordings of superimposed I_h_ before and during exposure to PGE1 (100 nM). Right: Mean instantaneous I_h_ (arrow) was plotted with respect to voltage and a linear regression was performed. Arrow indicates reversal potential of I_h_. Results are presented as the mean ± SEM.

**Figure 4 ijms-22-13534-f004:**
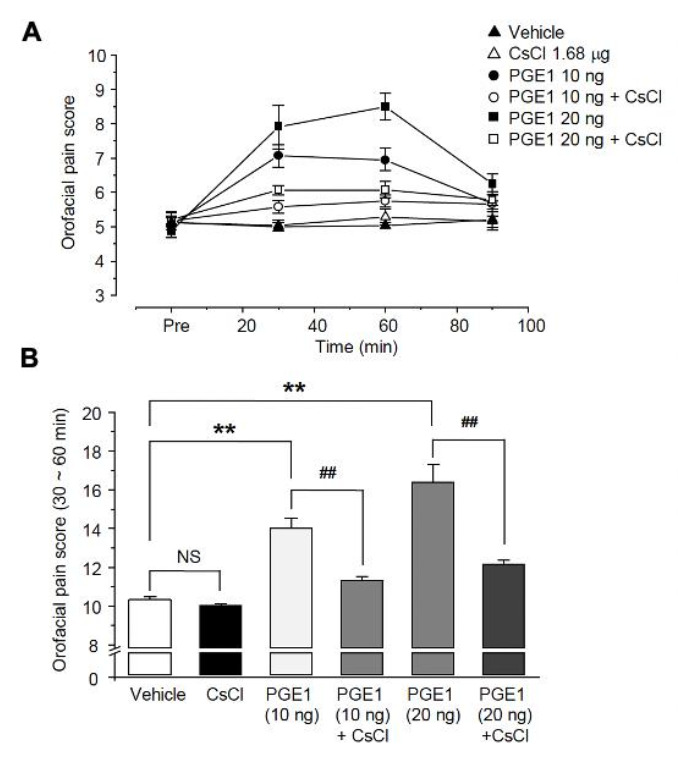
Role of HCN channels in PGER1-induced pain. (**A**) The orofacial pain score from a representative pain behavior for 90 min after the application of the vehicle, PGE1 (10 and 20 ng/10 μL), and a combination of PGE1 and CsCl (1.68 μg/10 μL) into the whisker pad of mice. PGE1 induced the mechanical allodynia in a dose-dependent manner (10 and 20 ng), and normal behavior was observed after 90 min. The observed PGE1-induced mechanical allodynia was blocked by CsCl, HCN blocker. (**B**) Summary of the orofacial pain scores measured for 30–60 min in each group. Results are presented as the mean ± SEM (** *p* < 0.001 compared to the vehicle group, ^##^
*p* < 0.001 compared to the PGE1 group, one-way ANOVA, NS: not significant).

**Figure 5 ijms-22-13534-f005:**
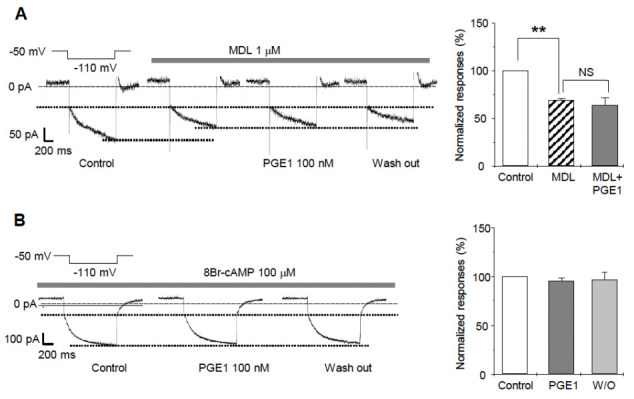
The effects of adenylyl cyclase inhibitor (MDL12330A) and 8Br-cAMP on PGE1-induced I_h_ facilitation. (**A**) A representative recording of I_h_ current after application of PGE1 (100 nM) in the presence of adenylyl cyclase inhibitor (MDL-12330A hydrochloride, 1 µM). Summary for the effects of PGE1 on I_h_ in the presence of MDL. I_h_ was significantly inhibited by MDL and was not facilitated by PGE1 after the application of MDL. (**B**) A representative recording of I_h_ after application of PGE1 (100 nM) in the presence of 8-Br-cAMP in the pipette solution. Summary for the effects of PGE1 on I_h_ in the presence of 8-Br-cAMP. Results are presented as the mean ± SEM (** *p* < 0.001, paired *t*-test). MDL: MDL-12330A, NS; not significant.

**Figure 6 ijms-22-13534-f006:**
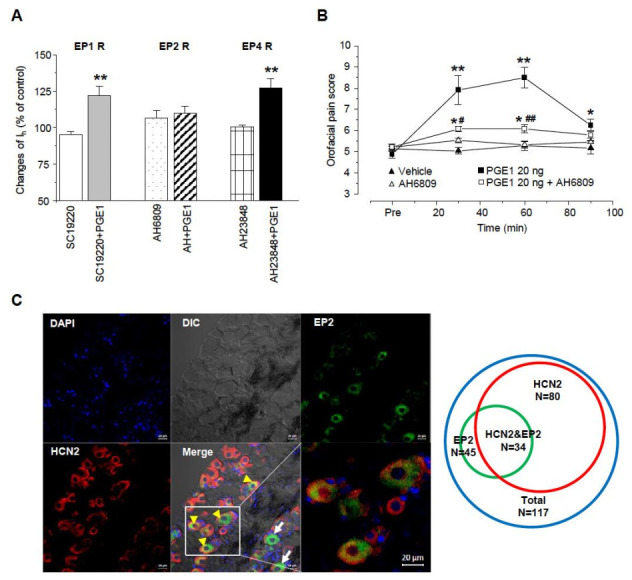
E-type prostaglandin (EP)-receptor of PGE1 for facilitation of I_h_. (**A**) Summary for the effects of PGE1 on I_h_ current in the presence of each EP receptor (EP1, EP2, and EP4) antagonist. EP2 receptor antagonist (AH6809, 10 μM) inhibited the facilitation of I_h_ by PGE1 (100 nM). SC19220: EP1 receptor antagonist, AH6809: EP2 receptor antagonist, AH23848: EP4 receptor antagonist. (** *p* < 0.001, paired t-test) (**B**) The orofacial pain score from a representative pain behavior for 90 min after the application of the vehicle, PGE1 (20 ng/10 μL), and a combination of PGE1 and AH6809 (10 μg/10 μL) into the whisker pad of mice. Results are presented as the mean ± SEM (* *p* < 0.05. ** *p* < 0.001 compared to the vehicle group; # *p* < 0.05. ## *p* < 0.001 compared to the PGE1 group, one-way ANOVA). (**C**) Left: Double immunostaining of EP2 with HCN2 channels in TG neurons. DAPI was used as tissue counterstaining. In TG, EP2 was present as the green color and HCN2 channels were present as red color on different interference contrast (DIC) images. Yellow arrowhead co-expression of EP2 and HCN2, arrow; expression of EP2, Bar = 20 μm. Right: Diagram of TG neuron population expressing HCN2 and EP2. Total TG neurons were represented by blue circle (n = 117), EP2 by green circle (n = 45), and HCN2 by red circle (n = 80), respectively. HCN2&EP2 refers to TG neurons (n = 34) in which both HCN2 and EP2 are expressed.

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
