# Peer review of "The Role of Prostaglandin E1 as a Pain Mediator through Facilitation of Hyperpolarization-Activated Cyclic Nucleotide-Gated Channel 2 via the EP2 Receptor in Trigeminal Ganglion Neurons of Mice"

_ijms, 2021, doi:10.3390/ijms222413534_

Round 1

Reviewer 1 Report

The authors present a well designed research investigating the role of PGE1 in the development of hypersensitivity in the orofacial region in mice and its mechanism of action. The methods are appropriate and clearly described. Results are clearly presented and supports the conclusions.

Author Response

We are grateful for your valuable comments on our manuscript.

Reviewer 2 Report

The study is aimed to investigate whether PGE1 induces pain behavior and how it acts on the hyperpolarization-activated cyclic nucleotide-gated 2 channel of TG neurons in mice.     The title is “The role of prostaglandin E1 as a pain mediator through facilitation of hyperpolarization-activated cyclic nucleotide-gated channel 2 via the EP2 receptor in trigeminal ganglion neurons of mice”.

  1. This is an animal study.
  2. Several factors influence the outcome of the study. Please discuss these.
  3. Please add the limitations of the study in the discussion section.
  4. What is the new knowledge of the article?
  5. Please recommend to the readers “How to apply this knowledge?”.

Author Response

We are grateful for your valuable comments that served to improve the quality of our manuscript. We have revised the manuscript and clarified the issues, which have been highlighted in yellow in the manuscript.

Reviewer 3 Report

The paper by Kwon et al, covers a very interesting topic with the activation of HCN channels through the EP2 receptor. I only have a few comments:

Is there any way you could specify or give more details on your neurons? For example, line 82 it just says “neurons” do you have anything like size etc.?

Line 100. The excitability was determined by HCN channels… I cannot follow this sentence, are you referring to a previous study [13], then could be “has been shown earlier to be determined by”

Line 106 TO eliminate iKir in the presence of BaCL2, is also not clear. Do you mean 1) IKir was eliminated by applying BaCl2, or 2) You eliminated IKir in the presence of BaCl2, and in that case did you add something more?

Line182 cAMP could activate HCN channels. Since you refer to [18,19], I assume that you mean it has been shown? By using could you imply this as an hypothesis not something that has been determined

Line 190, These results suggest that… and that cAMP-sensitive HCN2 channels are a subtype of HCN… By starting the sentence with suggest, what you then write is that the results suggest that HCN2 is a subtype of HCN channels, which is a fact. I suggest something like this: “and that the HCN2 channels subtype is the HCN channel most relevant in TG”

Line 208 what do you man by “we used each specific BP receptor antagonist”? please specify

Line 209 mediating that decrease in cAMP, should be “leading to a decrease in cAMP”

Line 279 activate HCN channels as its target molecule. This suggest that HCN is a molecule, I suggest removing molecule and ending the sentence with target. That would be sufficient.

Line 372 How can you anesthetize with an overdose of isoflurane? Why is that needed for decapitation?

Line 411, why do you fix in PFA overnight? This usually would be too long, and the antibody binding cites might be hidden, I suggest for the future to try a shorter protocol, e.g. 4-6 hours.

Author Response

(The authors gave the same response as above.)
